# Overview: Polycarbonates via Ring-Opening Polymerization, Differences between Six- and Five-Membered Cyclic Carbonates: Inspiration for Green Alternatives

**DOI:** 10.3390/polym14102031

**Published:** 2022-05-16

**Authors:** Zaher Abdel Baki, Hanna Dib, Tuba Sahin

**Affiliations:** College of Engineering and Technology, American University of the Middle East, Kuwait; hanna.dib@aum.edu.kw (H.D.); tuba.sahin@aum.edu.kw (T.S.)

**Keywords:** six- and five-membered cyclic carbonates, glycerol, ring-opening polymerization mechanism, ROP catalysis, green chemistry

## Abstract

This review aims to cover the topic of polycarbonate synthesis via ring-opening polymerization (ROP) of cyclic carbonates. We report a wide variety of ROP-initiating systems along with their detailed mechanisms. We focus on the challenges of preparing the polymers; the precise control of the properties of the materials, including molecular weight; the compositions of the copolymers and their structural characteristics. There is no one approach that works for all scales in cyclic carbonates ROP. A green process to produce polycarbonates is a luring challenge in terms of CO_2_ utilization and the targeted domains for application. The main resolution seems to be the use of controlled incorporation of functional/reactive groups into polymer chains that can tailor the physicochemical and biological properties of the polymer matrices, producing what appears to be an unlimited field of applications. Glycerol carbonate (GC) is prepared from renewable glycerol and considered as a CO_2_ fixation agent resulting in GC compound. This family of five-membered cyclic carbonates has attracted the attention of researchers as potential monomers for the synthesis of polycarbonates (PCs). This cyclic carbonate group presents a strong alternative to Bisphenol A (BPA), which is used mainly as a monomer for the production of polycarbonate and a precursor of epoxy resins. As of December 2016, BPA is listed as a substance of very high concern (SVHC) under the REACH regulation. In 2006, Mouloungui et al. reported the synthesis and oligomerization of GCs. The importance of GCs goes beyond their carbonate ring and their physical properties (high boiling point, high flash point, low volatility, high electrical conductivity) because they also contain a hydroxyl group. The latter offers the possibility of producing oligo and/or polycarbonate compounds that have hydroxyl groups that can potentially lead to different reaction mechanisms and the production of new classes of polycarbonates with a wide range of applications.

## 1. Introduction

Polycarbonates (PCs) are polymers with carbonate groups (-O-CO-O-) in their main chain. They have been classified as biodegradable [1] and biocompatible polymers [2], and they can be produced from renewable substances [3]. Thus, their use offers the possibility of developing sustainable technologies that are economically and ecologically attractive [4]. To date, polyesters have held the leading status in the field of polymers. However, PCs provide polydiester functionality, and their lack of acidic compounds that are related to biodegradation gives them a significant advantage over polyesters. In addition, the degradation rate of PCs is slower than that of polyesters. Due to all of these promising advantages, PCs have attracted great interest [5] concerning their potential application in many areas.

PCs have attracted the most attention over the past two decades for their significant potential in environmental and biomedical applications [6] due to their biocompatibility, biodegradability, bioresorbable activity [7], and the weak inflammatory properties of the degraded products [8,9]. There are several important biomedical uses of biodegradable polymers, some of which are listed below:Packaging is potentially the largest use of biodegradable polymers [10]. Biodegradable polymeric materials and blends that have passed toxicological and cytocompatibility testing for in vivo use have obvious benefits as biodegradable materials for pharmaceutical products, drugs, dressings for wounds, and surgical fixation (sutures, clips, bone pins, and plates). Biodegradable implants can be used as alternatives to metal implants for the fixation of fractured bones and joints [11].Controlled drug delivery may be the most important and versatile application of these polymers [12]. It has applications in the medical, veterinary, and agrochemical fields. Active ingredients, ranging from pesticides to contraceptives, can be delivered by sustained release followed by the ultimate biodegradation of the carrier medium.

Moreover, there are applications for PCs in the automobile industry as well as in lighting fixtures, textiles, microelectronic components, data storage, and construction-materials areas [13]. They also are used in the plastics industry because of their high impact resistance and thermal stability, due in part to the work of Nagahata et al. [14,15], by the solid-state, ring-opening polymerization of macrocyclic carbonates, which resulted in ultrahigh-molecular-weight bisphenol A-based polycarbonate (BPA-based polycarbonate) (MW > 2000 KDa). These polycarbonates are fabricated from BPA, and they have been in commercial use by Bayer and General Electric since 1958 [16]. However, because their biodegradation liberates the BPA, the World Health Organization (WHO) and Food and Drug Administration (FDA) [17] have expressed some concerns about the potential effects of BPA on the brain, human behavior, and the prostate gland in fetuses, infants, and young children. Therefore, FDA has declared that it supports the industry’s actions to stop producing BPA-containing baby bottles and infant feeding cups, baby formula packages, whereas the European Chemicals Agency (ECHA) has banned the use of BPA in thermal paper starting from January 2020.

Glycerol is one of the top 10 building blocks driven from biomass. Glycerol carbonate presents a multifunctional derivative of glycerol with a wide range of interesting properties, hence areas of applications. GC can be produced by integrating CO_2_ into glycerol, opening a new opportunity for carbon capture and utilization. Another point that makes glycerol carbonate more attractive is that some of its properties are placed between those of the cyclic alkylene carbonates (ethylene and propylene carbonate) and those of glycerol. Glycerol carbonate has a high boiling point (351 C), a low melting point (66.7 C), and a high dipole moment (l = 5.4 D) and a high dielectric constant value (e = 109.7) relative to many other organic compounds. Glycerol carbonate is also nontoxic and can be used as a polymer stabilizer and as a synthetic intermediate in organic reactions.

Additionally, at the polycarbonate-production level, polycondensation of cyclic carbonate has several challenging aspects, including poor control over the molecular parameters, limitation of molecular weights, and the use of toxic phosgene and pyridine [18]. The process also involves the phosgenation of hydroxyl compounds with KOH or pyridine as a catalyst or the transesterification of diols with lower dialkyl carbonates in the presence of a basic catalyst.

Considering the importance of greener routes for polycarbonates synthesis and the gravity of some related hazards, this review is focusing on the chemistry of the ring-opening polymerization of cyclic carbonates, presenting the advancements and the setbacks found in the literature. Overviewing the existing initiating systems with a deep comprehension of the associated mechanism shall incite the exploration of novel pathways in the field of polycarbonate industry.

## 2. Synthesis of Polycarbonates and ROP

Polycarbonates are divided into aliphatic polycarbonates (APCs) and aromatic polycarbonates (ArPCs), and the APCs are generally synthesized by either polycondensation or ring-opening polymerization. Polycondensation is performed by using activated carbonate derivatives with diphenols or diols [19] via a step-growth mechanism. Ring-opening polymerization (ROP) of cyclic carbonates proceeds in the presence of a catalyst and/or an initiator via a chain-growth mechanism that includes chain initiation and chain propagation; in the ideal case, there is no termination reaction. ROP offers the possibility of controlling the molecular weight, the distribution of molecular weights, the microstructure of the polymer, and the nature of its end groups. The importance of ROP has led us to explore all the ROPs associated with 5- and 6-membered cyclic carbonates for the preparation of polycarbonates.

The present review outlines the different synthesis pathways of polycarbonates, with the goal of investigating green synthesis routes for producing polycarbonates from bioresourced monomers. We focused on 5- and 6-membered cyclic carbonates as the “case study” monomers; the mechanisms of anionic, coordination-insertion, and cationic polymerization, and we presented the corresponding initiating systems. 

### 2.1. Kinetic and Thermodynamic Aspects of Cyclic Carbonates Ring-Opening Polymerization

The ROP of cyclic carbonates depends on both kinetic and thermodynamic factors, and it also depends on the reaction mechanisms associated with different catalyst/initiator systems.

The ROP mechanism of cyclic carbonates includes two steps [20], i.e., (1) initiation by means of ionic- (anionic or cationic) and insertion-type initiators and (2) regulation of the chain-growth mechanism by the equilibrium between kinetically controlled products and thermodynamically controlled products (Figure 1). 

In a spontaneous ROP, the Gibbs free energy must be negative (Δ*Gp* < 0), i.e., the enthalpy of ring-opening polymerization, Δ*Hp*, must be negative (Equation (1)), where Δ*S* is the polymerization entropy.
(1)ΔGp=ΔHp−TΔSp 

#### 2.1.1. Kinetic Control

In the regime of kinetic control, the chain propagation reaction is predominant, and a high-molecular-weight polymer (*P_n_*) is formed via Equation (2).
(2)$$nM\to P_{n+1}\;(M:\;\mathrm{monomer})$$

The polymerization conditions, such as solvent, temperature, concentration of the monomer, and especially, the nature of the active site must be adjusted to enhance the kinetically controlled regime of the reaction (where the polymer is formed) over the thermodynamically controlled regime (where a ring-chain equilibrium and the most probable chain-length distribution is established by means of transesterification reactions). For example, with Li^+^ and K^+^ as counter ions, the polymerization rates are much greater than the rates for Al^3+^ and Zn^2+^ in the active site [21].

Olsen et al. [22] have recently developed a switchable ROP of functional 6-member cyclic carbonate and ε-caprolactone monomers (i.e., AOMEC and εCL) catalyzed by TBD (1,5,7-triazabicyclo[4.4.0]-dec-5-ene). The synthesized nine-block copolymer was obtained in only 2.5 h. The switchable property was achieved by alteration of temperature during polymerization from +30 to −40 °C. A monomer-specific kinetic response arises for polymerization of εCL and retardation of the AOMEC polymerization.

#### 2.1.2. Thermodynamic Control

In the regime of thermodynamic control, back-biting reactions (loss of CO_2_ from carbonate to give an ether linkage) occur to form a low-molecular-weight fraction of cyclic oligomers (M_x_, Figure 1).

The monomer-polymer equilibrium is established in the kinetically controlled regime (Equation (3)):(3)$$P_{n}+M↔P_{n+1}$$

This equilibrium is characterized by equilibrium constant K (Equation (4)):(4)$$K=\left[P_{n+1}\right]/\left[P_{n}\right]{\left[M\right]}_{e}\approx 1/{\left[M\right]}_{e}$$

K is approximately equal to the inverse of the equilibrium concentration of the monomer ([*M*]*_e_*). The relationship between the equilibrium concentration of the monomer and the free enthalpy of polymerization is presented by Equation (5):(5)$$\ln{\left[M\right]}_{e}=\Delta H/RT-\Delta S/R$$

For standard conditions and above a critical temperature, no polymer is obtained. The critical temperature is defined as the ceiling temperature (T_c_ = Δ*H*/Δ*S*) when both Δ*H°* and Δ*S°* are negative.

In parallel, another equilibrium is established in a thermodynamically controlled regime. Ring-chain equilibrium (Equation (6)) is attained in function of K_x_, where K_x_ (K_x_ = [P_n−x_][M_x_]/[P_n_] ≈ [M_x_]_e_) is the equilibrium constant that determines the concentration of the cyclic oligomer with degree of polymerization x.
(6)$$P_{n}↔P_{n-x}+M_{x}$$

The equilibrium constant of an oligomer (K_x_) (K_x_ = [P_n−x_][M_x_]/[P_n_] ≈ [M_x_]_e_) is approximately equal to the equilibrium concentration of the oligomer ([M_x_]_e_), and K_x_ is proportional to the degree of polymerization to the −2.5 power [23,24].

In the thermodynamically controlled regime, further intermolecular transesterification reaction occurs, and the selectivity of the reaction and the microstructure of the polymer (in the case of copolymerization) depend on the ratios K_p_/K_b_ and K_p_/K_te_.

K_p_: rate constant of propagationK_b_: rate constant of back-bitingK_te_: rate constant of transesterification

These rate constants [25] are determined by the monomer that is used and by the nature of the active sites. It is of significant importance to understand the kinetic and thermodynamic aspects of ROP, as well as its different mechanisms.

### 2.2. Mechanistic Aspects

#### 2.2.1. Anionic Polymerization

The anionic ROP behavior of the cyclic carbonates depends on the size of their rings. Six- or seven-membered cycles tend to polymerize smoothly to yield the corresponding polycarbonates at lower temperatures (<100 °C) by anionic initiators. In contrast, the anionic ROP of the 5-membered ring is thermodynamically unfavorable and proceeds at higher temperatures (>150 °C, ethylene carbonate), causing elimination of carbon dioxide to give a copolymer that consists of both carbonate and ether linkages [26]. Anionic polymerization has been investigated in various solvents, such as tetrahydrofuran (THF), dimethylformamide (DMF), and toluene. In addition, bulk polymerization has been reported and the presence of a solvent in addition to the temperature at which the polymerization occurs enhances kinetic products over thermodynamic products. Anionic initiation of polymerization consists of a nucleophilic attack (Figure 2) of the initiator at the carbon atom of the carbonyl of the carbonate moiety, followed by an acyl-oxygen cleavage. The formation of active species, such as the alkoxide anion (alcoholate), was demonstrated by means of ^1^H and ^13^C NMR [25].

The nucleophilicity of the initiator and the electron affinity of the monomer control the efficiency of the initiation step. In the chain-growth reaction, the nucleophilic attack is repeated until the conversion of the monomer reaches a certain percentage.

#### 2.2.2. Coordination-Insertion Mechanism

In the insertion-coordination mechanism, different types of initiators can be used to start polymerization. However, detailed studies of the mechanism for cyclic esters have been conducted [27]; the findings in this research [28,29] will be extended to the field of polycarbonates with Kricheldorf et al. [30] who reported the ring-opening polymerization of trimethylene carbonate (TMC) with the same initiator used with cyclic esters, i.e., the tin octanoate (Sn-oct_2_) in chlorobenzene. They proposed the following mechanism for the initiation and propagation of the chain of the polymer (Figure 3)

In the step (**1**), they proposed a coordination and insertion of the TMC into the Sn-oct_2_ bond, followed by step (**2**), decarboxylation of the mixed anhydride that was formed, to produce an octanoate ester end group, while the chain growth continues as shown in step (**3**) via the normal coordination-insertion mechanism involving the Sn-alkoxide group. This mechanism was only proved for reactions that were conducted at temperatures of 140, 160, and 180 °C. The presence of a small amount of water or 1,3 diol as impurities with Sn-oct_2_ leads to a one-step, insertion-coordination mechanism instead of two steps (Figure 4), because water or alcohol is needed as a coinitiator for a sufficiently rapid polymerization that incorporates an alkyl (R in case of alcohol ROH) or a hydrogen atom (H in case of water) into the chain and forms carbonate end group (in the case of alcohol) [30]. This mechanism occurs at reaction temperatures that range between 100 and 120 °C.

Höcker et al. reported that a special initiator with an aluminum alcoholate group (Figure 5) is that tetraphenylporphyrin aluminum. This initiator also is known as the ‘Inoue catalyst’, and it is active for various monomers and is used for the preparation of block copolymers with one or two polycarbonate blocks [21]. Initiation occurs by the nucleophilic addition of alkoxide-aluminum-tetraphenylporphyrin (RO-Al-TPP) to the carbonyl group of the carbonate. The functional characteristics of this catalyst are of great interest because it is responsible for the initiation step of the ROP, and it also prohibits the backbiting by reintegrating the CO_2_ that is liberated during the ring opening back into the polymeric chain.

#### 2.2.3. Cationic Polymerization

Under cationic conditions, polymerization is accompanied by partial elimination of carbon dioxide, producing polycarbonates with an ether unit [31]; this mechanism of partial loss also was observed during the ring-opening polymerization of cyclic sulfite, during which sulfur dioxide was eliminated [32]. A cationic ROP of thiocarbonate with no loss was reported by Kameshima [33], and this led to additional investigation of the role of the catalyst. It was found that decarboxylation depends on the nature of the counter-ion present in the initiator [34]. Ariga et al. [35] proved that halide counter ions ensure cationic polymerization that suppresses the elimination of carbon dioxide. It is also important to control the reaction kinetics such that the initiation step is not faster than the propagation step; otherwise, the concentration of the monomer (TMC in Figure 1) would not be high enough to ensure a sufficient nucleophilic attack on the activated monomers (**1** or **2** in Figure 1) [36].

In Figure 1, two possible pathways are explained for the cationic polymerization of trimethylene carbonate initiated with methyl triflate (A^+^Y^−^ = MeOTf):Use of excess methyl triflate leads to formation of **(1)** where the counter-ion is covalently bonded to the initiated monomer, a nucleophilic attack on the covalent counter-ion adduct results in **(3)**, and this attack gives an intermediate oxycarbonyl cation **(4)**.Formation of **(2)** via alkylation of the exocyclic oxygen atom of the carbonate linkage, generating a trioxocarbenium ion, which undergoes a nucleophilic attack from another monomer to give adduct **(4)**.

Adduct **(4)** liberates carbon dioxide through the propagation steps to produce a polycarbonate **(5)** with ether units in its main chain. When alkyl halide was used as the initiator, a polycarbonate (Mn = 3000–5000) with a halide end group and no ether units was obtained.

#### 2.2.4. Challenge of Decarboxylation (Back-Biting)

The inconvenience of ionic polymerization is the decarboxylation (loss of CO_2_). Usually this is due to side reaction competing with propagation reaction or simply due to high reaction temperature. The advantage of the insertion-coordination mechanism is the absence of backbiting even at high temperatures (up to 180 °C) [36].

The decarboxylation from the carbonate group in the polymer chain can be both inter- and intramolecular, since it is always provoked by the oxide anion found at the end of the chain or that belongs to another polymer chain. Ariga et al. [35] studied the decarboxylation during chain propagation. They calculated the enthalpy associated with the formation of intermediate cations and proved that decarboxylation from the monomer is thermodynamically and kinetically favorable over that from the linear carbonate or polymer. On the other hand, Lee et al. studied the polymerization of ethylene carbonate, and they proposed a mechanism [37] (Figure 6) for the degradation of the polymer via decarboxylation.

The oxide anion takes a hydrogen atom, and then an electron-pair rearrangement occurs that gives a vinyl moiety (detected by ^1^H NMR) and a hydroxyl moiety. The other part of the polymeric unity loses a CO_2_ molecule to give an oxide anion at the end of the chain, which, in turn, can form a low-molecular-weight fraction of cyclic oligomers or ether-linkage units in the chain of the polymer. In summary the decarboxylation is occurring on several levels: starting with monomer initiation, continuing along chain propagation, and during chain degradation.

### 2.3. Cyclic Carbonate Monomers

One of the most interesting aspects of cyclic carbonates as monomers for ROP is their volume expansion aspect, since volume shrinkage during the polymerization process is a fundamental problem in the field of material science.

Takata and Endo [24,38] reported that cyclic carbonates polymerize with a considerable expansion in volume from 1.1 to 7.7% in contrast to common monomers that undergo volume shrinkage upon polymerization (methylmethacrylate or propylene oxide, for example). This was related to the overall decrease in density, since the decrease in density produced by the loss of intermolecular interaction caused by polymerization is more important than the increase in density due to the conversion of the monomer during polymerization (Figure 7).

The most common commercially available five- and six-membered cyclic carbonate monomers polymerized via ring-opening polymerization are listed below (Figure 8):

The TMC and DTC polymerizes faster than EC and PC due to less ring strain and thermodynamic conflicts. However, ROP of these monomers varies along with the applied catalyst-initiator system, leading to different mechanisms and products.

## 3. Catalyst-Initiator Systems in ROP of Six-Membered Cyclic Carbonates

The polymerization of six-membered cyclic carbonate (Figure 2) was first reported in the 1930s by Carothers et al., and it was obtained by heating trimethylene carbonate with a small amount of K_2_CO_3_, which was used as the initiator [39]. The molecular weights of the polycarbonates did not exceed 4 kilodalton (KDa). Later, other initiation systems, including organometallic and organic components, were investigated.

### 3.1. Organometallic Initiating Systems

Organometallic initiating systems are widely used in the technical production of polylactides, copolyesters of lactic acid, and the preparation of aliphatic polycarbonates. The high efficiency of these systems prompted several research groups to study the catalytic activity toward ring-opening polymerization of several monomers

#### 3.1.1. Main Group Metals and Transition Metals

Kricheldorf et al. [30] were the first to report the ring-opening polymerization of TMC initiated with Sn-oct_2_. The use of this catalyst was also reported later by other researchers [41,42,43,44]. The temperature range of 100–120 °C seemed to have an important role for the initiation mechanism. At lower temperatures, a coinitiator of R-OH type was needed for sufficiently rapid polymerization. This coinitiator was incorporated into the chain and formed carbonate end groups. Both initiation and propagation followed the pattern of the normal coordination-insertion mechanism. At temperatures above 120 °C, the TMC can undergo direct insertion into the Sn-Oct bond, thus polyTMC chains that had one octanoate ester end group per chain were formed. Furthermore, the molecular weights (Mn) can be controlled via the monomer/initiator ratio, and high-molecular-weight polymers can be obtained with short time reactions at 160 °C.

Darensbourg et al. [40,45,46] reported the use of biocompatible metals (such as Mg, Ca, Fe, and Zn) with obvious advantages over the use of tin complexes [47]. Their work focused on Schiff base derivatives known as Salen complexes (Figure 9). Since M(II) salen complexes do not possess internal nucleophiles for the chain initiation step, the presence of an anion initiator (cocatalyst) derived from n-Bu_4_N^+^Cl^−^ (tetrabutylammonium chloride) or PPN^+^N_3_^−^ [*µ*-nitrido-bis(triphenylphosphine)] is necessary to form an effective catalyst system for the ring-opening polymerization of trimethylene carbonate (TMC).

Their work demonstrated the catalytic activity of metals in the following order: Ca^2+^ > Mg^2+^ ≈ C_2_H_5_Al^2+^ > Zn^2+^, where the most effective catalytic system (with a TOF of 1286 h^−1^ at 86 °C) seemed to be Ca^2+^ complexed to a salen ligand with tert-butyl substituents in the 3,5-positions of the phenolate rings and an ethylene backbone for the diimine groups in addition to PPN^+^N_3_^−^ as a cocatalyst. Poly(TMC) was obtained with molecular weights ranging from 30 to 63 KDa and a narrow distribution band of molecular weights (1.48–1.76) [40].

Dobrzinsky [48,49] reported TMC and DMC polymerization with the use of acetylacetonates of low-toxic metals, i.e., iron, zinc, and zirconium. Zinc (II) acetylacetonate, Zn (acac)_2_, was proven to be a very good initiator of homopolymerization. The reaction proceeded via an insertion-coordination mechanism, with which almost complete conversion of the monomer (TMC) was observed after five minutes, and the polymers had high molecular weights (98.2–195.1 KDa) with a narrow distribution band of molecular weights (1.6–1.7). The reaction was conducted at 110 °C for a period ranging between 0.2 and 1.5 h, with a monomer conversion of 98% [48]. In correlation to the undesired backbiting, the study of the homopolymerization of six-membered cyclic carbonates and the copolymerization with cyclic esters using Li, K, Mg, Al, Zn, and Sn based catalysts, led to the conclusion that systems with alkali metals, i.e., Mg and Sn, were found to produce cyclic oligomers during polymerization, i.e., the rate of backbiting was of the same order of magnitude as the rate of polymerization. On the other hand, Al- and Zn-based catalysts resulted in ROP without cyclic oligomers formation [50], so the backbiting reaction rate was slower than the chain-growth reaction. The unavoidable inconvenience for this group of catalyst-initiators is that intramolecular and intermolecular transesterification reactions take place along with the polymerization reaction.

#### 3.1.2. Lanthanides

This group was studied thoroughly by Sheng et al. [51]. They worked with mixed-metal alkoxide clusters of lanthanide and sodium, such as [Ln_2_Na_8_(OCH_2_CH_2_NMe_2_)_12_(OH)_2_] and anionic lanthanide phenoxide complexes [52], such as [Ln(OAr)_4_][Na(DME)_3_]. These groups were found to be extremely active and presented a single-component initiator for the ring-opening polymerization of ε-caprolactone (ε-CL) and trimethylene carbonate. Polymerization proceeded through the cooperation of Ln and Na via a coordination-insertion mechanism to yield a high-molecular-weight polymer (23 × 10^4^ Da for ε-CL and 6.71 × 10^4^ Da for TMC) and a narrow distribution band of molecular weights (1.5–1.8 for poly(ε-caprolactone) and 1.6–2.2 for PTMC) taking into consideration that the reaction takes place at temperatures of 20 °C for ε-CL and 55 °C for TMC for short periods of three minutes for ε-CL and 30 min for TMC. Various research teams have studied and explored other lanthanide [53,54,55,56] complexes. The results shows a proven effectiveness for lanthanide complexes as promising initiators for the polymerization and copolymerization of TMC, DMC, and lactones [57,58,59]. The main issue facing the wide use of lanthanide complexes in ring-opening polymerization is the two required additional steps, i.e., (1) the preparation of these complexes since they are not commercially available; and (2) the purification process for the complexes [60,61].

#### 3.1.3. Lewis Acids

Lewis acids based on metal groups 2, 3, 4, 12, and 13, such as magnesium, calcium, scandium, yttrium, cerium, samarium, ytterbium, zirconium, zinc, aluminum, and tin triflates or triflimidates, in addition to rare-earth metal complexes [62,63,64,65,66,67], are known to be effective catalysts in combination with a proton source, such as an alcohol or a carboxylic acid, for the ROP of cyclic esters, such as ε-caprolactone and lactide. In general, using several bicomponent systems based on metallo-organic and inorganic catalysts is efficient in ring-opening polymerization, yet at different temperatures and with different mechanism and/or productivities [68]. The use of organometallic initiating systems in ring-opening polymerization has the advantage of a controlled polymerization of cyclic carbonate [69]. However, it has the problem of removing the metallic residues from the final polymer, which is especially important when the target application is drug delivery.

### 3.2. Organic Initiating Systems

Organic initiating systems initiate polymerization under mild conditions, thereby offering the possibility of obtaining high-molecular-weight polycarbonate without toxic initiators, such as heavy metal ions. Thus, such systems permit the synthesis of biodegradable or biocompatible polymers without toxic impurities. These initiating systems were based on:

#### 3.2.1. Brönsted Bases

The use of organic compounds such as amines [70] or guanidines [71] (Figure 10) in the copolymerization and immortal polymerization of cyclic carbonates was reported. Moreover, poly(ethylene glycol) [72,73] and natural amino acids also were used as catalysts for ROP of DTC and TMC [74]. The work of Nederberg [75] and Mindemark [76] represents a case study for the use of organic bases as catalysts for the ring-opening polymerization of TMC.

The polymers that were synthesized were isolated by precipitation; they had molecular weights up to 50 KDa and a polydispersity index (PDI) of 1.08. The presence or the absence of organocatalyst residues in the final product was not reported. Benzyl alcohol was used as the initiator in the reaction (Figure 11) to produce polymers with precise molecular weights and narrow distributions of molecular weights detected by gel-permeation chromatography (GPC). Moreover, the end groups were confirmed by ^1^H and ^13^C NMR.

In general, the catalytic activity of these base catalysts was exhibited towards the ROP of cyclic esters and carbonates. The guanidines, amidines, tertiary amines, N-heterocyclic carbenes, and thiourea tertiary amines proved similar activity in terms of monomer conversion percentage, but the highest DP, up to 420 KDa in the ring-opening polymerization of TMC, was only attained with the use of TBD [75]. This was due to the unique dual reception-donor functionality offered by the TBD catalyst; this dual function assures the activation of ROP without being directly inserted the mechanism of ROP through amide entity formation, as it is the case for the other amine-based catalysts [77].

The N-heterocyclic olefins (NHOs) are an emerging class of organopolymerization catalyst [78]. NHOs catalytic activity was investigated during the ROP of lactones and trimethylene carbonate (TMC). The reaction proceeds in the presence of Bn-OH at room temperature for a duration between 3.5 and 18 h; the conversion rates obtained did not exceed 76%, while the Mn of obtained polymers ranged between 5.8 kDa (for 3.5 h) and 7.2 kDa (for 18 h). Although the findings of these investigation places TBD as the first choice of catalyst, the findings related to the effect of polar moieties on the ring-opening polymerization could assist in developing tools for controlled polymerization.

#### 3.2.2. Brönsted Acids

The organic initiating systems can also be based on strong Brönsted acids that allow electrophilic activation of the monomer. Triflic acid (TfOH), methanesulfonic acid (MSA), trifluoroacetic acid (TFA) [31,38,79], Diphenyl Phosphate (DPP) [80], and imidodiphosphoric acid (IDPA) [81] have been reported as catalysts for the ring-opening polymerization of TMC, where alcohol or water was used as the initiator [82].

Matsuo et al. [79] reported the polymerization of six- and seven-membered cyclic carbonates, with TFA as a catalyst to promote the nucleophilic attack of alcohols. Less sterically hindered alcohols showed higher reactivity towards ring-opening polymerization, which produced polycarbonates (Mn = 2500–6800 Da) with hydroxyl end groups. Endo et al. [31] reported that ester-substituted, six-membered cyclic carbonates polymerize faster than six-membered cyclic carbonates. They used triflic acid as an initiator at a reaction temperature of 20 °C for a long period (24 h). This produced a polycarbonate/ether polymer with a carbonate-to-ether linkage ratio of 7/3, molecular weights Mn = 1800–5000 Da, and a narrow distribution of molecular weights of Mw/Mn = 1.10–1.29. Furthermore, Delcroix et al. [82] reported the ring-opening polymerization of TMC catalyzed by triflic acid and methanesulfonic acid, using alcohol or water as initiators in stoichiometric amounts. The MSA was found to have similar polymerization kinetics to TfOH, although it is less acidic. Therefore, activity does not simply correlate with acidity. Moreover, MSA did not induce decarboxylation, and a polyTMC ether linkage-free polymer was produced with a molecular weight Mn = 2350–5800 Da and a narrow distribution of molecular weights of Mw/Mn = 1.08–1.17. The use of IDPA led to formation of star-shaped polycarbonates [81] after 34 h reaction at room temperature. Alternating the benzyl alcohol with other polyols did not have much effect on the molar mass of the polymer (Mn ≈ 5 kDa), while the narrowest distribution was obtained with BnOH (Mw/Mn =1.13).

### 3.3. Enzymes

It is noteworthy that enzymatic ring-opening polymerization has been studied extensively for cyclic esters with lipase as the enzymatic initiator [83,84,85,86]. It was reported that this approach is applicable for certain cyclic carbonates (Figure 12).

An interesting result was obtained for the polymerization of DTC catalyzed with immobilized porcine pancreas lipase (IMPPL). When the polymerization proceeded at 120 °C for four days, poly(DTC) was obtained with a high molecular weight (15.7 KDa) and a narrow distribution band of molecular weights (PDI = 1.4) and the monomer conversion efficiency was 84.9% [87]. Better results were achieved using the second recycled IMPPL as catalyst, including higher molecular weight (26.2 KDa) and improved yield (97.6%) of polycarbonate.

## 4. Catalyst-Initiator Systems in ROP of Five-Membered Cyclic Carbonates

It is well known that five-membered alkylene carbonates undergo ROP with difficulty due to thermodynamic requirements [77]. Ethylene carbonate and propylene carbonate ROP have been studied extensively. Their polymerization occurs above 100 °C; it always results in poly (alkylene ether-carbonate) [88] polymer due to the loss of CO_2_ during the reaction. This decarboxylation affects the thermodynamic aspects of ROP by making the entropy positive, thus causing the polymerization to take place at high temperatures. Furthermore, the loss of CO_2_ during ROP depends on the mechanism imposed by the used initiating systems.

### 4.1. Main Group Metals and Transition Metals Initiating System

The reaction proceeds in the presence of metal alkoxide and metal acetylacetonates, as well as metal alkyls, as the catalyst. The optimized reaction temperature was found to be 170 °C for EC and 180 °C for PC. The homopolymerization for 5-membered cyclic carbonates was not possible since the polymerization is always accompanied by decarboxylation that leads to poly(alkylene ether-carbonate), and the loss of carbon dioxide during the polymerization exceeds 50 mol%; thus, the obtained copolymer content of carbonate units is less than 50 mol%.

The polymerization of five cyclic carbonates has been reported by several researchers. Among these is the significant work of Vogdanis et al. [88], on the ethylene carbonate monomer. They reported that the composition of poly(ether-carbonate) was strongly dependent on the catalyst that is chosen, and they used several catalysts, including dibutyltin dimethoxide, and butyllithium. They observed that CO_2_ retention decreased as the relative basicity of the catalyst increased.

The carbonate component of the product did not exceed 50 mol% in any experiment. Their explanation was that the polymerization occurred above the ceiling temperature (Tc) for the formation of pure poly(ethylene carbonate) and that propagation occurred via monomer insertion between the growing end of the polymer chain and a catalyst fragment (Figure 3).

Above Tc, equilibrium (1) must be shifted to the left since the sequence length of ethylene carbonate units cannot exceed one [89] Ethylene carbonate can be added when one carbon dioxide is lost. This equation shows the importance of the catalyst in CO_2_ retention (Equations (2) and (3), Figure 4).

A novel mechanistic vision was presented via NMR investigations during the reaction time in that the results indicated that there were two possible routes to react the monomer. In the first step, the growing polymer can attack the carbonyl carbon of the carbonate (Figure 5).

The other possible route would occur by attack at the alkylene carbon (Figure 6). Kinetically, the carbonyl attack is favored over the alkylene attack. However, the carbonyl attack is reversible. Thermodynamically, the cyclic monomer is favored over the polymer, and the attack on the alkylene carbon is irreversible and accompanied with CO_2_ loss; therefore, the most probable mechanism includes both an alkylene carbon attack and a carbonyl carbon attack.

### 4.2. Brönsted Bases Initiating Systems

A Brönsted base catalyst was also used to accelerate the polymerization reaction, but a significant loss of CO_2_ (carbonate units) was detected. Two parameters could be affected by this loss, i.e., (1) the polymerization reaction becomes more favorable from a thermodynamic point of view because the enthalpy of the reaction decreases as CO_2_ is lost; and (2) the molar fraction of carbonate units within the polymeric chain decreases (0.22–0.17).

The polymerization of ethylene carbonate with basic initiators has rarely been studied due to the basicity effect on CO_2_ during polymerization, but Lee et al. [37] studied the polymerization of ethylene carbonate (EC) initiated by KOH at various temperatures (150–200 °C) and ratios of carbonate to initiator (1000:1 to 20:1) and discovered that the reaction can be described as a two-step process. In this manner, the final polymer is the result of initiation and propagation as well as chain cleavage. In the first step, the molecular weight of the polymer increases with reaction time to a maximum in the range of 1000–9000 Da, depending on the carbonate-to-initiator ratio that is used. Generally, the polymer has a carbonate-to-ether linkage ratio of 2 (i.e., 30–32% carbonate), which remains fairly constant until approximately 90–100% of the monomer has been converted. In the second step, the molecular weight and carbonate content of the polymer decrease significantly with continuous heating.

Furthermore, ethylene carbonate and propylene carbonate undergo polymerization in the presence of bispenol A/KHCO_3_ as an initiator/catalyst system, as described by Kéki et al. [90] and by Soos et al. [91]. The EC formed polyether oligomers, while the polycarbonate-ether oligomers resulted from PC polymerization with limited yield. When KHCO_3_ was replaced by an organic base, as described by Wu et al. [92], polyether diol was formed as a result of PC polymerization. Conversely, when tert-butylphenol was used with KHCO_3_ in the co-oligomerization of EC, PC, and ε-caprolactone, the result was co-oligomers with *t*-butylphenol and OH head groups with two types of co-oligomers, i.e., cyclic and linear oligomers with low carbonate linkage compared to the co-oligomers without carbonate linkage [93].

### 4.3. Dual Initiating Systems-Lewis Base-Lewis Acid 

Lewis acid catalysts were never reported as a meaningful catalyst for EC polymerization. While Lewis base catalysts are not known to play an important role in accelerating the polymerization reaction of ethylene carbonate (reaction period between 72 and 96 h), but they were important for the retention of CO_2_, i.e., carbonate units, where the detected molar fraction of ethylene carbonate present in the polymeric chain was up to 40% [88].

Recently, a newly emerging approach employing dual catalytic polymerization catalysis was reported [94]. The catalytic activity towards EC polymerization of a combination of cooperatively acting Lewis bases ((including N-heterocyclic olefins, phosphazenes and nitrogen bases)) and Lewis acids (such as LiCl, MgF_2_, BEt_3_), has been proved. The polymerizations were conducted under microwave irradiation at T = 160–200 °C with low catalyst load ranges between 0.4–0.005 mol%, resulting polymers of molar masses up to 10,000 g/mol.

### 4.4. Glycerol Carbonate RO-Polymerization/Oligomerization

The ring-opening polymerization of glycerol carbonate has not been studied extensively. The GC monomer has so far been used to produce hyperbranched polyglycerol [95]. The polymerization of glycerol carbonate proceeds via anionic approach with simultaneous decarboxylation. A partially deprotonated trimethylolpropane (TMP) was used as an initiator. The product of this process is a hyperbranched polyether with pendant hydroxyl groups. The reaction occurs at 170 °C, over 12 h, with 10% (wt/wt) of initiator. The obtained oligomer attained an Mn up to 1000 Da.

A different approach with a focus on CO_2_ retention was studied and patented in the work of Mouloungui et al. [96]. The oligomerization of glycerol carbonate was achieved via a one-pot in situ approach. The starting reagents were glycerol, urea [3:2 molar ratio], and monohydrated zinc sulfate (1.3% wt/wt) was used as catalyst. The reaction proceeds at 140 °C for 3 h, and a pressure of 4 kPa. This step is meant to produce glycerol carbonate while eliminating the ammonia formed. The absence of ammonia flow would indicate the end of the reaction [97]. The reactional mixture includes the produced glycerol carbonate, excess glycerol, and the catalyst. Then, oligomerization reaction proceeds in situ by increasing the temperature to 160 °C for 150 min, at atmospheric pressure. This results in polyhydroxylated oligomers (Mw < 1000 Da) rich in linear carbonate groups.

In view of all the reported work of 5-membered cyclic carbonate ROP, the challenge remains controlling decarboxylation reaction that occurs in parallel with polymerization reaction. Attaining the highest level of carbonate retention is beneficial in terms of intensifying the desired properties of linear carbonates. Thus, it is of great interest to find a catalyst/initiator system that assures a maximum retention of CO_2_ units within the polymeric chain, since all of the previous work has reported the production of a heteropolymer that contains both carbonate and ether units. A promising answer may be found in the work of Inoue et al. [98,99,100] who reported the synthesis of aliphatic polycarbonate starting from an epoxide, while bubbling CO_2_ at 1 atm in the presence of aluminum tetraphenylporphyrine acetate as a catalyst at room temperature. The Inoue catalyst demonstrated a bifunctional activity that should be capable of initiating the ROP of- membered cyclic carbonate and reintegrating the released CO_2_ units into the polymeric chain.

## 5. Conclusions

An inclusive review for the initiating systems of ring-opening polymerization of cyclic carbonates has been reported. It is apparent that there is no ‘perfect’ system that is preferred for all applications. Significant differences exist between the polymerization of 5-membered and 6-membered cyclic carbonates. The challenge of having decarboxylation reaction along with the ROP could be avoided via the use of Inoue’s kind of catalyst, which can guarantee minimum decarboxylation at mild-pressure conditions. All the reported synthetic routes and results represent a source of inspiration for us; many applications that target the transport of biologically active substances through biological membranes require biosourced and biodegradable materials. One important parameter that has a key role in enhancing the mobility through biological membranes is the molar volume of the carrier (or the transporter of the active substance); the mobility of the carrier increases as its size decreases, and with this kind of linear relationship between molar volume and mobility, we should direct our focus towards development of new, oligomeric structures rather than polymeric structures. Seeking oligomers through ring-opening oligomerization would overcome the inconvenience of long reaction periods for 5-membered cyclic carbonate, and the glycerol carbonate monomers represent an attractive candidate that is compatible from both synthesis and application perspectives.

## Data Availability

Not applicable.

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
