# Peer review of "Overview: Polycarbonates via Ring-Opening Polymerization, Differences between Six- and Five-Membered Cyclic Carbonates: Inspiration for Green Alternatives"

_polymers, 2022, doi:10.3390/polym14102031_

Round 1

Reviewer 1 Report

This manuscript is a review for the initiating systems of ring opening polymerization of 5-membered and 6-membered cyclic carbonates along with their detailed mechanisms. The manuscript focuses on the challenges of preparing the polymers; the precise control of the properties of the materials, including molecular weight; the compositions of the copolymers and their structural characteristics. These results would be of potential interest to the readership of polymers. Furthermore, the manuscript is well presented, and the conclusions are reasonable. Therefore, I support the publication of the manuscript in polymers after minor revision.

  • I suggest adding “3.1.1 main group metals and transition metals” after 3.1 and changes 3.1.1 lanthanide to 3.1.2
  • Page 2 line 77 polyceondensation ?
  • The numbers (1), (2), (3), (4) and (5) of Scheme 1 should be bold.
  • The mechanisms of five-membered cyclic carbonate polymerization should be discussed in detail.

Author Response

We want to thank you for your time in assessing our work. We have considered your feedback and applied all the recommended modifications as mentioned below.

  • I suggest adding “3.1.1 main group metals and transition metals” after 3.1 and changes 3.1.1 lanthanide to 3.1.2. We applied the suggested modification
  • Page 2 line 77 polyceondensation ? the spelling mistake was corrected
  • The numbers (1), (2), (3), (4) and (5) of Scheme 1 should be bold. We applied the suggested modification
  • The mechanisms of five-membered cyclic carbonate polymerization should be discussed in detail. We applied the suggested modification.

Reviewer 2 Report

The review describes polymerization mechanism, and condition for ring-opening polymerization of six and five-membered cyclic carbonate. It focuses on the chemistry and mechanistic aspects of the polymerizations. However, it does not dive deep on discussion about green alternatives to commonly used monomers. Such discussion is currently lacking.

Both abstract and conclusions need to be rewritten to better describe the focus and scope of the paper, which as it stands is more about the chemistry and polymerization mechanism and conditions than green alternatives and biological applications. For instance, it is extremely odd how both the abstract and conclusion mention glycerol carbonate (GC) as a valid and interesting alternative to BPA, yet no discussion of this monomer can be found in text whatsoever. Only 1 cited paper mention the use of this monomer, but the use is not described at all in detail.

Additionally, all cited papers are from before 2010. I highly advise the authors conduct a more extensive search of the literature to include more recent papers and advancement in the field. As it stands, the current review is outdated and would not be as useful for the field. For instance, authors cite and report a table of conditions for the polymerization of ethylene carbonate taken from a paper from 1986. It would be much more relevant to prepare a table describing uses in the last decade or so.

Overall, the flow of the manuscript is good and easy to read, although it would also benefit from minor editing and English check.

As of now, I cannot recommend publication of this review. Publication can be reconsidered after major review.

More detailed comments and suggestions can be found below:

  • Ref 1. Only describe biodegradation of PCs, not biocompatibility.
  • Ref 4. Does not match the description. It is a very specific synthetic method to produce poly(trimethylene carbonate), which does not describe lines 44-47.
  • Ref 8. Describes applications in the biomedical field and not in any of the fields cited by the authors.
  • The introduction ending is too abrupt. Authors should consider adding a paragraph describing what the focus of the review will be about. For instance, I would move lines 91-95 here.
  • Can the author provide a couple more examples of kinetically controlled ROP reaction in section 2.1.1.?
  • In figure 2, move the parenthesis of the alcoholate intermediate so it does not overlap with the counterion X+.
  • Fig 3, It is unclear where the arrows are pointing and where they are coming from.
  • Ref 29. In the text (line 211) authors mention Kameshima as one of the authors, but the cited reference is not from that author and does not relate to thiocarbonate. Additionally, the references do not proceed in order, passing from ref 25 to ref 29.
  • Scheme 1. It would be nice to also show the cationic initiation step other than the propagation.
  • All the examples provided for the “mechanistics aspect” (section 2.2) are related to 6-membered rings. Can the authors also provide examples of 5-membered rings and outline possible differences in reaction mechanism or conditions?
  • Line 232 and 338. Mn reported is missing unit of measurement (Da?). Make sure they are reported throughout the text.
  • 5 and fig. 9. Define R groups.
  • Section 3.1.2 Most of cited reference describe ROP polymerization of lactones and lactide, and not carbonate. Find more relevant references.
  • Section 3.2.1. It would be useful to have some discussion about advantages and disadvantages of the different base catalyst.
  • Section 4 should have the same subsection structure of section 3 for consistency. Moreover, only a few papers are discussed here compared to the many of section 3.

Author Response

We would like to thank you for your time in assessing our work. We have taken into consideration your valuable feedback. Thanks to your comments we believe our review is more solid and beneficial now. We hope that you value the modifications we made, and we remain available for any further amelioration.

Please find below our replies to your comments, point by point:

Thank you again

---------------------------------------------------------------------------------------

Both abstract and conclusions need to be rewritten to better describe the focus and scope of the paper, which as it stands is more about the chemistry and polymerization mechanism and conditions than green alternatives and biological applications. For instance, it is extremely odd how both the abstract and conclusion mention glycerol carbonate (GC) as a valid and interesting alternative to BPA, yet no discussion of this monomer can be found in text whatsoever. Only 1 cited paper mention the use of this monomer, but the use is not described at all in detail.  

We added a paragraph about glycerol carbonate in the introduction and presented the reported GC polymerization in the section of 5-membered cyclic carbonate (section 4.4) 

Additionally, all cited papers are from before 2010. I highly advise the authors conduct a more extensive search of the literature to include more recent papers and advancement in the field. As it stands, the current review is outdated and would not be as useful for the field.  

We added and discussed more recent papers to the review  

For instance, authors cite and report a table of conditions for the polymerization of ethylene carbonate taken from a paper from 1986. It would be much more relevant to prepare a table describing uses in the last decade or so. We decided to remove the mentioned table and updated the concerned section.  

Overall, the flow of the manuscript is good and easy to read, although it would also benefit from minor editing and English check. 

As of now, I cannot recommend publication of this review. Publication can be reconsidered after major review. 

More detailed comments and suggestions can be found below: 

Ref 1. Only describe biodegradation of PCs, not biocompatibility.  

We added another reference related to biocompatibility

Ref 4. Does not match the description. It is a very specific synthetic method to produce poly(trimethylene carbonate), which does not describe lines 44-47.

We replaced it with two other references

Ref 8. Describes applications in the biomedical field and not in any of the fields cited by the authors.  

We replaced it with other references

The introduction ending is too abrupt. Authors should consider adding a paragraph describing what the focus of the review will be about. For instance, I would move lines 91-95 here.

We applied the recommended move of lines and developed the introduction part.  

Can the author provide a couple more examples of kinetically controlled ROP reaction in section 2.1.1.? We added another example detailing the effect of temperature   

In figure 2, move the parenthesis of the alcoholate intermediate so it does not overlap with the counterion X+. We modified the figure to avoid the overlapping 

Fig 3, It is unclear where the arrows are pointing and where they are coming from. 

We fixed the arrows to show the formation of the anhydride on one side and the insertion of Sn entity on the other side  

Ref 29. In the text (line 211) authors mention Kameshima as one of the authors, but the cited reference is not from that author and does not relate to thiocarbonate.  

We added Kameshima reference. The cited reference is related to thiocarbonate, specificaly in page 313.  

Additionally, the references do not proceed in order, passing from ref 25 to ref 29. We addressed this issue  

Scheme 1. It would be nice to also show the cationic initiation step other than the propagation. We added the initiation step to the scheme 

All the examples provided for the “mechanistics aspect” (section 2.2) are related to 6-membered rings. Can the authors also provide examples of 5-membered rings and outline possible differences in reaction mechanism or conditions? We answered this question throughout the text  

Line 232 and 338. Mn reported is missing unit of measurement (Da?). Make sure they are reported throughout the text. We added the missed units added and checked for all the others 

  • and fig. 9. Define R groups. We defined R groups in the mentioned figures

Section 3.1.2 Most of cited reference describe ROP polymerization of lactones and lactide, and not carbonate. Find more relevant references. We added more relevant references  

Section 3.2.1. It would be useful to have some discussion about advantages and disadvantages of the different base catalyst. We added the requested discussion 

  • Section 4 should have the same subsection structure of section 3 for consistency. Moreover, only a few papers are discussed here compared to the many of section 3.

The literature related to 6-membered cyclic carbonate is much more than that of 5-membered. We worked on section 4 to make it systematically consistent with section 3 although the content is different.

Round 2

Reviewer 2 Report

I thank the author for addressing my comments.

I believe the review is in a better shape now and I approve of its publication.